# Assessing Decision Support Tools for Mitigating Tail Biting in Pork Production: Current Progress and Future Directions

**DOI:** 10.3390/ani14020224

**Published:** 2024-01-10

**Authors:** Sophia A. Ward, John R. Pluske, Kate J. Plush, Jo M. Pluske, Charles V. Rikard-Bell

**Affiliations:** 1Australasian Pork Research Institute Ltd., Willaston, SA 5118, Australia; j.pluske@april.org.au (J.R.P.); c.rikardbell@april.org.au (C.V.R.-B.); 2Faculty of Science, The University of Melbourne, Parkville, VIC 3010, Australia; 3SunPork Group, Eagle Farm, QLD 4009, Australia; kate.plush@sunporkfarms.com.au; 4SciEcons Consulting, Perth, WA 6010, Australia

**Keywords:** decision support tools, tail biting, pigs, agriculture, technology

## Abstract

**Simple Summary:**

Tail biting in pigs is an abnormal event where one pig engages in the biting, chewing, or oral manipulation of another pig’s tail. The repeat biting of the wounded site can lead to pain, infection, and the possible mortality of the victim pig(s), which is why it is a serious issue in pork production. Tail biting is often difficult to prevent as there are various reasons why a particular pig may choose to exhibit this behavior. The aim of this review is to identify current decision support tools and other technological aids that can be used to predict the likelihood of a tail biting event. Additionally, we aim to understand how dependable these decision support tools are for predictive tail biting events by examining both the underlying model and data utilized for generating predictions.

**Abstract:**

Tail biting (TB) in pigs is a complex issue that can be caused by multiple factors, making it difficult to determine the exact etiology on a case-by-case basis. As such, it is often difficult to pinpoint the reason, or set of reasons, for TB events, Decision Support Tools (DSTs) can be used to identify possible risk factors of TB on farms and provide suitable courses of action. The aim of this review was to identify DSTs that could be used to predict the risk of TB behavior. Additionally, technologies that can be used to support DSTs, with monitoring and tracking the prevalence of TB behaviors, are reviewed. Using the PRISMA methodology to identify sources, the applied selection process found nine DSTs related to TB in pigs. All support tools relied on secondary information, either by way of the scientific literature or expert opinions, to determine risk factors for TB predictions. Only one DST was validated by external sources, seven were self-assessed by original developers, and one presented no evidence of validation. This analysis better understands the limitations of DSTs and highlights an opportunity for the development of DSTs that rely on objective data derived from the environment, animals, and humans simultaneously to predict TB risks. Moreover, an opportunity exists for the incorporation of monitoring technologies for TB detection into a DST.

## 1. Introduction

Tail biting (TB) in pigs is a serious issue with a potential to impact both the welfare of pigs and the economic viability of the pork industry [1]. The multifactorial and complex nature of TB makes it difficult to investigate [2,3]. Consequently, a common practice in production units is to routinely remove a portion of the pig’s tail at birth (docking) to prevent TB behaviors from occurring in later stages of production. Many papers and reviews have previously described the possible causes and risk factors linked to TB [4,5,6,7], for example, the mixing of unfamiliar pigs [2], the absence of enrichment [8,9], and reduced space [10,11]. Consequently, TB is typically seen as abnormal behavior and not as a reflection of normal social hierarchy. Hence, it is an adverse response to unfamiliar and/or stressful situations [12,13]. As farms can be set up with multiple production stages, pen layouts, and health statuses, it can be difficult to pinpoint the reasons for outbreaks of TB behavior; thus, mitigation is often difficult [5,12,13]. Within the constraints of model development, a Decision Support Tool (DST) can integrate on-farm welfare considerations, production system factors, and (or) cost implications associated with TB. As such, the use of a DST to assist managers, veterinarians, and stockpersons with decision making associated with either the prevention or consequences of TB outbreaks is a logical option.

For the purpose of this review, a DST is any interactive computer-based system intended to help identify, take a course of action, and solve problems, as described by Kukar et al. [14]. There are two main model types that are used to calibrate DST predictions, namely, mechanistic or empirical modeling [15]. Mechanistic models use physical and biochemical principles to predict a given outcome, such as the likely spread of disease in a unit, feed conversion rates, or nutrient utilization [16,17]. When a prediction needs to be made on the likely behavior of an animal, such as the feed intake of a sow or, in this case, the likelihood of a TB event, empirical modeling may be more suitable. Empirical modelling uses assumptions made to input variables to generate predictions [18], which is why it requires large amounts of data to determine these assumptions. For this type of DST to be reliable, one would expect the data to be sourced from a large range of appropriate parameters to ensure the risk assessments are true to the situation.

The overall aim of this paper is to determine whether any DSTs or related technologies have been developed to aid in identifying risks of TB events on a farm. As such, this review includes literature from both the developers of identified DSTs and any sources related to the validation or adoption of these tools with intended users. The validation of a model is defined as any study testing the accuracy of TB predictions either by sensitivity analyses, expert opinions, or empirical research [19]. Validation studies, along with an evaluation of DST methodologies and information sources, should give greater insights into the usefulness of these technological aids in identifying the likelihood of a TB event. In addition to DST tools and validation studies, monitoring technologies that can be used in conjunction with a risk assessment model are included in the review. Technologies that successfully capture the prevalence of TB events, and associated behaviors, can help in the further validation of support tools and improve the reliability of risk assessments.

## 2. Materials and Methods

A review of relevant databases was conducted in accordance with the Preferred Reporting Items for Systemic Reviews and Metanalysis (PRISMA) guidelines [20]. Searches were conducted on the Web of Science, PubMed, Scopus, and AGRICOLA, as these are known for their coverage of scientific and agricultural literature.

### 2.1. Database Keywords and Search Terms

Following the original research question, the aim of this review was to identify DSTs that could include technologies related to the use of these tools, to help mitigate TB in commercial pork production. As such, the search strings for all databases included the following key search terms: ‘Pig or Pigs or Swine’ and ‘Decision support Tool or Technology or Decision support system’ and ‘Tail biting’ or ‘Tail damage’ or ‘Tail posture’ or ‘Tail lesions’. No limitation was set on the language of publication to ensure relevant sources were included.

### 2.2. Inclusion and Exclusion Criteria

To account for advancements in technology in the 21st century, the sources included in this review were limited to those published in 2000–2023.

In alignment with the objectives of this review, only papers directly related to commercial pork production were included. For example, papers published in journals, such as Journal of Dairy Science, were excluded for their lack of relevance to the review subject.

Subsequently, the remaining sources underwent a title-based screening process to determine their relevance to the use of DSTs in mitigating TB. Only sources addressing the use of technologies to mitigate TB in pork production were retained for review.

The literature returned from the data search were filtered first for their relation to technology use in pork production and, second, for date of publication. This process resulted in a total of 1490 sources being selected out of an initial pool of 2104 for the subsequent phase of screening.

The objective of this review was to identify DSTs and (or) related technologies used in the mitigation of TB in pork production. Thus, when assessing these sources for their alignment to the primary research question, 1457 sources were deemed not relevant and were thus excluded from the review.

## 3. Results

Following this exclusion stage, the remaining 29 sources underwent full-text screening to determine their relevance to this review. This examination led to the identification of five relevant DSTs from the selected databases. These DSTs included PIGTAIL [21], Husbandry Advisory Tool [4], SchwIP [22], Tail biting cost simulator [6], and Tail biting risk simulator [23].

Additionally, two more DSTs, Welzijncheck-Varkens [21] and SAPARO [24], were discovered through the tail biting inventory complied by Professor Sabine Dippel (S. Dippel, personal communication, 5 July 2022). Using the keywords associated with these identified DSTs in a Google Scholar search, two additional DSTs, BEEP [25] and Tail Biting Risk Assessment Tool [26], were identified in the gray literature.

In addition to the DSTs, the review’s screening process also obtained 16 sources related to TB tracking technologies, six sources related to DST adoption for TB mitigation, and seven validation studies associated with the recognized DSTs.

A summary of this screening process is outlined in Figure 1.

### 3.1. Current TB Decision Support Systems

The screening process resulted in nine relevant DSTs related to the risk assessment of TB in pork production, as presented in Table 1. Empirical modeling was used for these DSTs, with assumptions of TB risk taken either from the relevant literature or using responses given by industry ‘experts’ (including stakeholders, veterinarians, and farm managers). Out of the nine DSTs identified, no model described the use of primary observational data to determine TB predictors in the regions of intended use.

In addition to identifying the risk factors, seven of the nine DSTs employed secondary information to assess the significance, or weighting, of each factor in influencing the risk of TB. The other two DSTs collated factors associated with TB risk but did not incorporate any weighting scores into the model (WØ). How the risk factors were used to predict the risk of a TB event varied between DSTs, with the assessment falling into one of four main categories. Models either simulated the likelihood of a TB event by running simulations on a given scenario (simulation output); by collating risk factors into a single report (risk factors identified); by presenting risk factors in order of TB influence (classification predictive); or by assessing the risk variables using a regression analysis and generating a numerical score (regression predictive). A summary of each tool, their model, and source of information are shown in Table 1.

### 3.2. Tool Validation Review

Since these DSTs relied on secondary information that could be subjective, it was important to examine tool validation studies. Out of the sources identified, only five DSTs had evidence of model validation, with SchwIP being the most externally tested DST. Nearly all studies used primary on-farm data to determine tool predictability, with the exception of PIGTAIL using cases in the literature to assess TB risk outputs [22]. A summary of the validation studies, the DSTs tested, and main conclusions are presented in Table 2.

### 3.3. Monitoring Technologies to Assist DSTs

Table 3 presents the various tracking technologies applicable for capturing and monitoring TB behaviors. The monitoring technologies included manual applications [24], the use of video surveillance to capture TB behaviors [30,31,32,33,34,35,36,37], or the individual monitoring of pig behaviors using physical Radio-Frequency Identification (RFID) ear tags [38,39,40,41,42]. Additionally, environmental sensors provided warnings of environmental TB risks by monitoring the concentration of ammonia [43,44] in a shed. With the improvements to TB tracking algorithms and connection to neural networks [31,33,34,37], camera tracking proved to be the most versatile tracking technology for identifying the risks associated with TB. This included the identification of a TB event, feeding behavior, engagement with enrichment, and environmental risks.

## 4. Discussion

### 4.1. DST Information Sources

All the DSTs identified in this review used secondary sources, either by way of the literature or responses obtained from relevant stakeholders, to determine the likely causation for TB behavior in their models. Notably, the *Husbandry Advisory Tool*, *SchwIP*, and *PIGTAIL* used this information to also rank the influence of each risk factor, incorporating weighting scores into their empirical models. As previously discussed, empirical models rely on assumptions to predict the likelihood of a given event. Consequently, it is important to evaluate how these DSTs rely on secondary sources to generate model predictions. To predict TB events on a given farm, *PIGTAIL* collated and ranked the variables from 97 publications that contained relevant information on TB. These sources ranged in time (1966–2001) and geographical location (The Netherlands, UK, Denmark, Ireland, and Canada) without accounting for variations that could exist between environmental or animal differences.

It is important to factor the geographical location of the sourced data, as management and production practices can vary. The DSTs *HAT* and *SchwIP* used responses from relevant ‘experts’ in pig production to generate the model predictions. Although these responses were useful for identifying the risk factors likely associated with a TB event, they risked being subjective in nature. As an example, Madey et al. [31] noted similarities in the type of risk factors identified in *SchwIP* and *HAT*, yet the ranking of each risk factor for influencing a TB event varied between the tools. It was suggested that the UK-based support tool *HAT* ranked ‘provision of enrichment’ as a greater influence for mitigating TB behavior over *SchwIP* due to UK experts being more experienced with straw-based systems. When it came to developing a model for TB predictions, Scollo et al. [5] suggested that any DST needed to consider the various interactions that could exist between the animal, management, and the ever-changing physical surroundings in a particular unit. This approach is partially adopted by *HAT*, where the user can choose between a system reliant on straw or a straw-free system to evaluate the risk of other TB factors [4]. Constructing a model utilizing primary observational data, such as the regression analysis developed by Scollo et al. [5], can help capture variable interactions for better predictive outputs.

### 4.2. DST Validation

Given the sporadic nature of TB events, validating these DSTs requires observations to be performed over a long period of time, ideally across seasons and production settings to avoid confounding results. Evidently, the number of validation studies is low, with validations for only five of the nine identified DSTs. Grumpel et al. [22] suggested that a true validation study for TB needed to assess nearly all the pens on a farm/unit of interest, as TB outbreaks are not a frequent event. When SchwIP predictions were tested on farms, vom Brocke et al. [28] noted a significantly higher prevalence of TB lesions in the first three months of the study, with less events for the remainder of the year. It was unclear whether the reduction in TB prevalence was due to the ongoing implementation of SchwIP or the result of animals being less susceptible to TB risk factors in the second part of the year. When testing a DST, it is preferable to assess predictions over a longer period to account for seasonal and/or animal variations between batches.

When evaluating the validation studies, it is also important to consider when TB lesions are assessed. When evaluating the effectiveness of SchwIP, vom Brocke et al. [28] only assessed pigs for TB lesions after they were processed at the abattoir. Without a universal method for validating these support tools, it is difficult to compare their models and predictive outputs.

In addition to a universal validation method, it is important for these DSTs to be evaluated in the region of intended use, as the reasons for biting behavior are shown to vary across geographical locations and genotypes. For instance, SchwIP was modeled off HAT to suit German farm conditions [22] and BEEP was modeled off SchwIP to suit French production systems [25]. It was therefore difficult to compare TB DSTs in one validation study, as each model was developed to best suit a particular region. The Tail Biting Cost Simulator and PIGTAIL both account for possible variations in herd genotypes, with one of the questions being whether pigs are mixed with a breed of pig found to exhibit more biting behavior than other breeds (Landrace). Considering the heritability of aggressive behavior can be more complex than whether animals have Landrace genetics, the DST Welzijncheck-Varkens questioned the user on boar and sow lines in the herd before presenting the questionnaire. Having preliminary questions on herd-specific information can help with addressing the interactions that can exist between a given group of animals and improving the predictions in an existing DST.

Another consideration when validating these DSTs is the extent to which the suggested modifications provided by a TB risk output are put into practice. When validating HAT, Taylor et al. [4] found that only the farms that received renumerations for implementing the suggested changes showed a reduction in TB lesions over time. If strategies to reduce TB are not implemented properly on a farm, it is difficult to validate model predictions.

### 4.3. Current Attitudes and Issues Regarding the Use of DSTs

An important aspect of the adoption of a DST is to assess users’ attitudes towards the technology and practicality/feasibility of the suggested strategies. Vom Brocke et al. [28] noted usability as being the leading factor of interest for stakeholders when considering the adoption of SchwIP. Similarly, producers in The Netherlands agreed there was a need for computerized systems, but were concerned with the complexity of the tool, particularly if their stockpersons were the intended users. If stockpersons were expected to navigate a DST themselves, the length of time required to input the data was a factor of DST adoption, especially on larger units with multiple pen types and shed layouts [47]. To assist with manually recording TB prevalence on farms, SAPARO developed an iOS/Android application to tally the severity and position of tails in each pen. These data were then sent to the SAPARO DST interface to allow the user to track the success of the changes suggested by the DST over time [24].

Another concern regarding adoption is the perceived usefulness of a DST, particularly if the suggested changes are difficult or not feasible to implement [46]. Producers expressed that ‘increasing space allowance’ to reduce TB risk was not straightforward with finite space being available [48]. The ‘provision of enrichment’ can also be a difficult strategy to implement due to the differences in the shed design and layout and the type of enrichment that can be used [48,49]. Although straw provision is a well-studied strategy for reducing TB risk [10], it may not be practical on farms with slatted flooring due to potential effluent blockages [4,49,50]. If other enrichment is provided, it is important to clarify the features of the type used, as non-chewable materials can have little effect on pigs for reducing TB risk [10]. Rather than simply identifying the risks, it is beneficial for a DST that considers variable interactions to provide alternative strategies in the report. In addition to the validation of current DST models, studies focusing on the adoption of these tools, their user friendliness, and application on farms can help with the development of future models.

Incorporating features of TB DSTs together can help improve tool usability and overall tool usefulness. The Tail Biting Cost Simulator does not provide strategies for reducing TB risk but can simulate the economic impact of a perceived outbreak across a unit [6]. This type of model predictor is a useful addition to a DST that identifies the possible risks so that any suggested investments in farm management can be justified.

### 4.4. Tracking Technologies to Aid DST Use

Given the financial investment and time taken to implement a DST’s recommended changes, the integration of tracking technologies to monitor TB events can allow for the better management of TB-related risks. In addition to the web-based tool SAPARO, an iOS/Android application can be used to record the prevalence of TB lesions, their severity, and the position of tails in each group of pigs [24]. The data are then sent back to the DST dashboard allowing a farm manager to monitor the prevalence of TB over time [24]. Although a helpful tool for tracking biting on smaller units, it can be more suitable to set up an automatic tracking system in larger commercial farms. The results of this review show that most research on TB tracking is in the area of computer vision technology over physical RFID tagging. Although RFID tagging has been shown to be a successful monitoring tool [39,40,42], there are limitations to the types of behaviors that can be tracked with a corresponding RFID reader [41]. Camera tracking technologies have the potential to track whatever the computer model is set to capture, meaning direct TB perpetrators [37] and their victims [30] can be detected before a serious outbreak occurs. This information can also provide DSTs with more reliable data and so can also help in improving predictive risk outputs. In these cases, the use of an automatic tracking system for TB and associated behaviors can be more suitable. Ultimately, the use of a tracking technology (or technologies) has the potential to work alongside a DST to assist with the monitoring of TB risk factors.

## 5. Conclusions

Improving pig welfare and farm profitability are key factors for improving sustainability in the pork industry. This review of the current literature revealed nine DSTs related to evaluating the risks of TB on farms. These tools currently rely on either subjective information from production experts or from the results of various studies on TB in the literature. Given the subjective nature of these prediction models, it is important to show evidence of reliability through validation studies. This review found a limited number of studies related to the validation of these support tools, with *SchwIP* presenting the greatest number of studies. Consequently, there is a need for the further validation of these DSTs in regions of intended interest, ideally with the study accounting for seasonal variations and differences in production stages. This is more feasible with the ongoing improvements to TB tracking technologies, enabling DST-managed farms to monitor TB risks.

In addition, this review highlighted the potential for a novel DST model that relied on objectively collated environmental-, animal-, and human-related data to not only identify TB-related risk factors, but the interactions between variables that could influence the risks. Moreover, behavioral, and physiological indicators, based on measurable parameters, can be integrated into a DST model to assess pig welfare. The implementation and validation of such a tool, possibly in conjunction with real-time tracking technologies, can offer valuable assistance in managing and reducing TB risk on farms.

## Figures and Tables

**Figure 1 animals-14-00224-f001:**
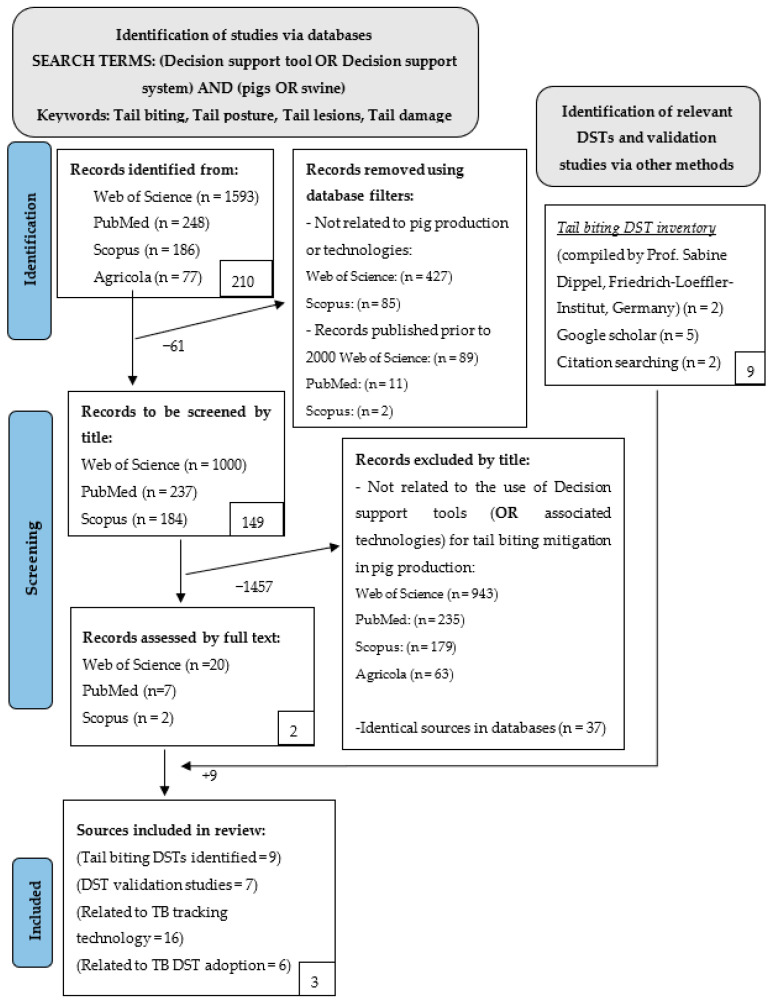
Flow diagram showing the steps of methodology and selection processes.

**Table 1 animals-14-00224-t001:** Summary table for TB DSTs, their developers, models, and information sources. Key—W1: risk factors assigned a weighting score; WØ: no weighting score.

Primary Developer	DSTs	Model	Information Source for DST Model Predictions
Wageningen University and Research Centre, The Netherlands	PIGTAIL	Regression predictive (W1)	Literature
	Welzijncheck-Varkens	Risk factors identified (W1)	Expert Opinion
	Tail biting simulator	Simulation (W1)	Expert Opinion
Agricultural and Horticultural Development Board, UK	Husbandry Advisory Tool (HAT)	Classification predictive (W1)	Literature; Expert Opinion
Friedrich Loeffler Institute, Germany	SchwIP	Regression predictive (W1)	Expert Opinion; Based on HAT
IFIP Institute Du Porc, France	BEEP	Regression predictive (W1)	Expert Opinion; Based on SchwIP
University of Helsinki, Finland	SAPARO	Risk factors identified (WØ)	Expert Opinion
Facility of Veterinary Medicine, Utrecht University	Tail biting risk assessment tool	Risk factors identified (WØ)	Literature
Natural Resources Institute, Finland	Tail biting outbreak cost simulator	Simulation (WØ)	Expert Opinion

**Table 2 animals-14-00224-t002:** Overview of validation studies associated with the identified TB DSTs.

DST	Validation Source	Study Conclusions
PIGTAIL	Literature	PIGTAIL model predictions corresponded to 67 out 72 study outcomes [26]
SchwIP	Primary observational data	There was a correlation between higher SchwIP risk scores and increased prevalence of TB on German farms [27]
SchwIP	Primary observational data	The application of SchwIP strategies on farm reduced TB lesions, but only after three months into the validation study [28]
SchwIP	Primary observational data	Implementing suggestions made by SchwIP to reduce TB risk did not significantly prevent TB in conventionally managed, undocked herds [29]
Welzijncheck-Varkens	Primary observational data	DST needs to clarify what an acceptable level is for each risk factor in the report so the right changes can be made to mitigate TB behaviors [29]
Husbandry advisory tool	Primary observational data	Farms were given suggestions for reducing TB risks, but only farms that were also given a financial incentive to implement the said changes observed a reduction in the TB risk over time [4]
Tail biting risk assessment tool	Primary observational data	Validation determined the tool to not be suitable yet for use by livestock farmers/veterinarians [26]

**Table 3 animals-14-00224-t003:** Technologies that could be used to assist TB DSTs by monitoring the presence of biting behavior and/or related risk factors.

Technology Description	Overview	Measurements
Application compatible with iOS and Android devices	Able to manually record signs of tail lesions and tail posture in pig pens. Reports graphically presented to show TB prevalence over time	Tail posture; TB lesions [24]
Radio-Frequency Identification (RFID) ear tags	Monitors individual pigs (feeding, movement, activity); time dependent; requires linking to appropriate software	Feeding [38,39]; Activity [40]; Enrichment interaction [42]
Camera technology	Application of algorithms to detect behaviors relating to a TB outbreak using video monitors	Tail posture [30] TB events [37] Feeding behavior [33] Enrichment interaction [33,34] Environmental risks [43]
Environmental sensors	Monitoring adverse environmental conditions that can increase the risk of TB behaviors	Air quality [44,45,46]

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
