# Peer review of "Assessing Decision Support Tools for Mitigating Tail Biting in Pork Production: Current Progress and Future Directions"

_animals, 2024, doi:10.3390/ani14020224_

Round 1

Reviewer 1 Report

Comments and Suggestions for Authors

Dear Authors, I have carefully reviewed this manuscript. The paper provides valuable insights into Decision Support Tools for Mitigating Tail Biting. However, I have identified some areas that may require revisions.

1. I appreciate the use of meta-analysis for the literature search in the study. However, to enhance reproducibility and transparency, it is essential that you provide the search terms used for each database in their methodology section. This information is crucial for readers who may wish to replicate your results.

2. I would like to acknowledge that my expertise in the fields of Animal Behavior and Decision Support Tools is limited. I believe that the use of the four databases mentioned in the manuscript is adequate. However, it appears that the search terms ((Decision support tool or Decision support system) AND (pigs or swine)) used in the literature review may be somewhat limited. I would recommend considering the inclusion 'Tail Biting' or other search terms to broaden the scope of the search.

3. I would suggest merging rows with the same Primary Developer, such as rows 1, 7, and 8, which appear to be identical. Besides, there are only two models in the table, so I recommend to merge and classify them as well.

4. In table 1, the row 8 "L: O" should be "L; O" ?

Author Response

Thank you for your time editing this paper. With amendments to the review, we are now over the appropriate word count required for this type of paper. Please see our comments to your edits. 

Thankyou

Reviewer 1:

Dear Authors, I have carefully reviewed this manuscript. The paper provides valuable insights into Decision Support Tools for Mitigating Tail Biting. However, I have identified some areas that may require revisions.

Thankyou for your time reviewing this paper and offering your constructive points.

  1. I appreciate the use of meta-analysis for the literature search in the study. However, to enhance reproducibility and transparency, it is essential that you provide the search terms used for each database in their methodology section. This information is crucial for readers who may wish to replicate your results.

Thank you for this feedback. We agree that including specific search terms for each database will help increase the reproducibility of our findings. Lines 77 – 108 provide a more specific outline of the review process.

  1. I would like to acknowledge that my expertise in the fields of Animal Behavior and Decision Support Tools is limited. I believe that the use of the four databases mentioned in the manuscript is adequate.

We have included a comment regarding database selection to provide clarity at lines 80-81. In addition, we included AGRICOLA as a database due to its specificity to agricultural papers. Although this database was included to increase the number of sources retained for this review, no unique papers were identified.

Lines 70 – 72:

Reviews and Meta Analysis guidelines (PRISMA) [20]. Searches were conducted in the Web of Science, PubMed, and Scopus, as these are known for their coverage of scientific and agricultural literature.

However, it appears that the search terms ((Decision support tool or Decision support system) AND (pigs or swine)) used in the literature review may be somewhat limited. I would recommend considering the inclusion 'Tail Biting' or other search terms to broaden the scope of the search.

To follow this comment, we included ‘Tail biting’, ‘Tail-biting’, ‘Tail posture’ and ‘Tail damage’ as relevant keywords (now included in Figure 1). Although ‘tail biting’ would broaden the scope of the search, the aim of this review was to determine decision support tools (and/or relevant technologies) that were used to mitigate the issue of tail biting in pig production. As such, papers that relate to tail biting but not technologies would not be included in this paper.

  1. I would suggest merging rows with the same Primary Developer, such as rows 1, 7, and 8, which appear to be identical. Besides, there are only two models in the table, so I recommend to merge and classify them as well.

DSTs that were developed with the same primary developer have been merged.

  1. In table 1, the row 8 "L: O" should be "L; O" ?

Amended

Reviewer 2 Report

Comments and Suggestions for Authors

The review, titled “Assessing Decision Support Tools for Mitigating Tail Biting in 2 Pork Production: Current Progress and Future Direction“ addresses an important and timely topic. I found the subject matter of the article fascinating and read the manuscript with great interest. The paper aligns well with the scope of the journal. However, I believe that in its current form, it has several shortcomings.

This paper focuses on the use of Decision Support Tools (DSTs) to predict and mitigate tail biting (TB) behavior in pigs, a complex issue influenced by various factors. The paper reviews existing DSTs and their methodologies for assessing TB risk on farms. It highlights the limited external validation of these tools and emphasizes the need for DSTs that rely on objective data from the environment, animals, and humans to predict TB risks. This review contributes to a better understanding of DST limitations and underscores the potential for developing more robust and data-driven DSTs for TB prevention in pig farming.

There are some areas that need consideration:

One major limitation highlighted in the review is the lack of external validation for most DSTs. This raises questions about their reliability and effectiveness in real-world farm settings. Future research should prioritize the validation of DSTs to ensure their practical utility.

The review points out that the existing DSTs rely heavily on secondary information, such as scientific literature and expert opinion. Developing DSTs that incorporate objective data from the farm environment, animals, and human factors would enhance their accuracy and relevance.

While the paper discusses the potential economic benefits of using DSTs to prevent TB, it could delve deeper into the cost-effectiveness of implementing these tools on farms. Farmers often need to assess the economic viability of adopting new technologies or practices.

TB in pigs can be influenced by human factors, such as management practices and farm worker behavior. Future DSTs should consider incorporating human-related variables to provide a more comprehensive analysis.

The paper could explore the practical challenges and barriers that farmers may face when implementing DSTs. It's crucial to address how user-friendly these tools are and whether they require specialized training.

Tail biting prevention should also consider the ethical treatment of animals. The paper could touch upon how DSTs align with animal welfare standards and guidelines.

Specific comments:

I suggest rewriting the simple summary. According to the author's guidelines, this section should summarize and contextualize your paper within the existing literature in your field. It should be written without technical language or nonstandard acronyms, with the goal of conveying the meaning and importance of this research to non-experts.

Introduction: I recommend that the authors enhance the introduction by providing a more comprehensive literature review on various aspects of pig welfare throughout the production cycle. This can include:

Rearing Conditions: Discussing studies and findings related to housing systems, space requirements, and environmental enrichment in pig rearing. Highlighting the impact of housing on pig behavior and overall well-being. (https://doi.org/10.3390/app13179731)

Feeding Practices: Exploring research on feeding strategies, nutrition, and its effects on pig health and behavior. This can encompass discussions on dietary preferences, feeding enrichment, and the importance of balanced nutrition. (https://doi.org/10.3390/su15064814)

Transportation: Addressing the welfare concerns associated with pig transportation, including factors like transport duration, loading, unloading, and environmental conditions during transit. (10.3390/ani10122386 and 10.3390/ani10060945)

Slaughter Practices: Discussing the various methods of pig slaughter, their impact on animal welfare, and the importance of humane slaughter practices. (https://doi.org/10.1080/1828051X.2023.2212004)

Pain Management: Highlighting studies related to pain management in pig farming, especially during procedures like castration, tail docking, and teeth clipping. Emphasizing the importance of pain mitigation. (https://doi.org/10.1371/journal.pone.0284218)

Behavioral and Physiological Indicators: Discussing the use of behavioral and physiological indicators as tools for assessing pig welfare. This can include stress-related behaviors, cortisol levels, and other measurable parameters. (https://doi.org/10.3389/fnbeh.2023.1173298)

By incorporating a broader literature review in the introduction, the authors can provide a more holistic context for their research on Decision Support Tools (DSTs) for tail biting prevention. This will help readers better understand the significance of DSTs in the broader context of pig welfare and production.

It's important to ensure that the Material and Methods section includes relevant references to provide a clear understanding of the methodology used in the study. I recommend that the authors consider incorporating the suggested reference (10.3390/educsci12080573) or any other appropriate references that are relevant to the methods employed in their review. This will help readers better grasp the methodological framework and its alignment with established practices or innovations in the field.

Author Response

Thank you for your time editing this paper. With amendments to the review, we are now over the appropriate word count required for this type of paper. Please see our responses to your comments below. 

Thankyou 

Reviewer 2:

The review, titled “Assessing Decision Support Tools for Mitigating Tail Biting in 2 Pork Production: Current Progress and Future Direction “addresses an important and timely topic. I found the subject matter of the article fascinating and read the manuscript with great interest. The paper aligns well with the scope of the journal. However, I believe that in its current form, it has several shortcomings.

Thank you for your time reviewing this paper and offering your constructive points.

This paper focuses on the use of Decision Support Tools (DSTs) to predict and mitigate tail biting (TB) behavior in pigs, a complex issue influenced by various factors. The paper reviews existing DSTs and their methodologies for assessing TB risk on farms. It highlights the limited external validation of these tools and emphasizes the need for DSTs that rely on objective data from the environment, animals, and humans to predict TB risks. This review contributes to a better understanding of DST limitations and underscores the potential for developing more robust and data-driven DSTs for TB prevention in pig farming.

There are some areas that need consideration:

One major limitation highlighted in the review is the lack of external validation for most DSTs. This raises questions about their reliability and effectiveness in real-world farm settings. Future research should prioritize the validation of DSTs to ensure their practical utility.

Yes, this is something that we wish to highlight as a finding in this review. See lines 339-345.

The review points out that the existing DSTs rely heavily on secondary information, such as scientific literature and expert opinion. Developing DSTs that incorporate objective data from the farm environment, animals, and human factors would enhance their accuracy and relevance.

While the paper discusses the potential economic benefits of using DSTs to prevent TB, it could delve deeper into the cost-effectiveness of implementing these tools on farms. Farmers often need to assess the economic viability of adopting new technologies or practices.

We agree that the economic viability is a primary driver for adopting new technologies or practices of adoption. However, we didn’t find evidence of relevant analyses in this review.

TB in pigs can be influenced by human factors, such as management practices and farm worker behavior. Future DSTs should consider incorporating human-related variables to provide a more comprehensive analysis.

Agreed, there is potential for a DST to be developed with the use of observational data on animal, environment, and human related variables in the region of intended interest and have refined the conclusion as such. See lines 343-349.

The paper could explore the practical challenges and barriers that farmers may face when implementing DSTs. It's crucial to address how user-friendly these tools are and whether they require specialized training.’

Agreed, in addition to more tool validation studies, there is a need for more workshopping with producers and understanding their attitudes towards adoption of current DSTs available. Section 4.3 has been altered to better address this issue.  

Tail biting prevention should also consider the ethical treatment of animals. The paper could touch upon how DSTs align with animal welfare standards and guidelines.

Specific comments:

I suggest rewriting the simple summary. According to the author's guidelines, this section should summarize and contextualize your paper within the existing literature in your field. It should be written without technical language or nonstandard acronyms, with the goal of conveying the meaning and importance of this research to non-experts.

Simple summary has been rewritten to be clearer to the reader.

Introduction: I recommend that the authors enhance the introduction by providing a more comprehensive literature review on various aspects of pig welfare throughout the production cycle. This can include:

Rearing Conditions: Discussing studies and findings related to housing systems, space requirements, and environmental enrichment in pig rearing. Highlighting the impact of housing on pig behavior and overall well-being. (https://doi.org/10.3390/app13179731)

Feeding Practices: Exploring research on feeding strategies, nutrition, and its effects on pig health and behavior. This can encompass discussions on dietary preferences, feeding enrichment, and the importance of balanced nutrition. (https://doi.org/10.3390/su15064814)

Transportation: Addressing the welfare concerns associated with pig transportation, including factors like transport duration, loading, unloading, and environmental conditions during transit. (10.3390/ani10122386 and 10.3390/ani10060945)

Slaughter Practices: Discussing the various methods of pig slaughter, their impact on animal welfare, and the importance of humane slaughter practices. (https://doi.org/10.1080/1828051X.2023.2212004)

Pain Management: Highlighting studies related to pain management in pig farming, especially during procedures like castration, tail docking, and teeth clipping. Emphasizing the importance of pain mitigation. (https://doi.org/10.1371/journal.pone.0284218)

Behavioral and Physiological Indicators: Discussing the use of behavioral and physiological indicators as tools for assessing pig welfare. This can include stress-related behaviors, cortisol levels, and other measurable parameters. (https://doi.org/10.3389/fnbeh.2023.1173298)

By incorporating a broader literature review in the introduction, the authors can provide a more holistic context for their research on Decision Support Tools (DSTs) for tail biting prevention. This will help readers better understand the significance of DSTs in the broader context of pig welfare and production.

This recommendation is appreciated, and we have expanded the introduction to emphasize the importance of welfare. However, we are confident that the developers of the DST’s have incorporated welfare attributes wherever possible. Hence, by maintaining the current focus we believe that this review offers a comprehensive examination of the current technology available for assessing TB risk. Nevertheless, we have reiterated the importance of welfare considerations in our conclusions.

It's important to ensure that the Material and Methods section includes relevant references to provide a clear understanding of the methodology used in the study. I recommend that the authors consider incorporating the suggested reference (10.3390/educsci12080573) or any other appropriate references that are relevant to the methods employed in their review. This will help readers better grasp the methodological framework and its alignment with established practices or innovations in the field.

 Thank you for this feedback. We agree that including specific search terms for each database will help increase the reproducibility of our findings. Figure 1 has been updated with exclusion and inclusion stages for more specific review process.    

Round 2

Reviewer 2 Report

Comments and Suggestions for Authors

Reviewing the introduction the important part of the transportation is still missing, please include it. Addressing the welfare concerns associated with pig transportation, including factors like transport duration, loading, unloading, and environmental conditions during transit. (10.3390/ani10122386 and 10.3390/ani10060945).

Moreover, I recommend incorporating discussions in your paper regarding the dissemination of your results to the broader public through modern platforms like social media. This proactive approach is essential to counteract the prevalence of misinformation within the field of animal science.

In light of the ubiquity of fake news, particularly in the realm of animal science, emphasizing the need to share your findings through contemporary channels such as social media becomes paramount. Consider elaborating on the potential impact of your research on public understanding and awareness. Discuss how leveraging platforms like Twitter, Facebook, Instagram or other social media outlets can contribute to the accurate dissemination of information, reaching a wider audience beyond the academic and scientific community.

Please see: 10.3390/ani13223503

Highlighting the importance of transparent communication and engaging the public directly through accessible channels will not only enhance the credibility of your research but also foster a better-informed society. By embracing modern communication tools, you can actively combat the spread of misinformation and ensure that your valuable contributions make a meaningful impact in both academic and public spheres.

Author Response

I think we have received feedback intended for another paper. The suggested papers mentioned (10.3390/ani10122386 and 10.3390/ani10060945) are not related to tail biting or associated technologies.

Could we please clarify whether any further edits are required  

Thankyou 

Sophie